# Effects of Genotype, Direct Sowing and Plant Spacing on Field Performance of *Jatropha curcas* L.

**Zafitsara Tantely Andrianirina [1], Matthias Martin [2,\*], Euloge Dongmeza [3] and Elisa Senger [2]**

1   TatsAina Agro Consulting, Lot III R 49 Tsarafaritra Tsimbazaza, Antananarivo 101, Madagascar
2   JatroSolutions GmbH, Echterdinger Str. 30, 70599 Stuttgart, Germany
3   JatroSahel SARL, Yaoundé B.P. 14155, Efoulan Yaoundé, Cameroon
\*   Correspondence: Matthias.Martin@JatroSolutions.com

**Abstract:** The tropical multiuse tree *Jatropha curcas* L. (jatropha) is highly promoted as oilseed crop for biodiesel production and for climate change mitigation, but cultivation practices require further research. The objectives of this study were to assess the effects of varying plant spacings (2.0 m × 4 m compared to 1.5 m × 4 m), crop establishment methods (raising plantlets in a nursery prior to planting to the field compared to direct sowing) and genotypes on seed yield, seed quality and plant height, recorded at a dry-subhumid location in Madagascar (Ihosy) and at a humid location in Cameroon (Batchenga). Averaged across treatment variants and genotypes, seed yield and seed oil content were higher at the dry-subhumid site and in particular the narrower spacing reached higher seed yields per unit area than the wider spacing. At the humid site, plant growth was characterized by strong accumulation of biomass. The establishment method tested at the dry-subhumid site showed no significant differences in the recorded parameters. Our results encourage to re-think common practices in jatropha cultivation and underpin the importance of the correct choice of location, genotype and agronomic practices considering the interactions between all factors.

**Keywords:** jatropha; genotypes; direct sowing; plant spacing; agro-technology; agronomy; Madagascar; Cameroon

## 1. Introduction

*Jatropha curcas* L. (jatropha) is a tropical multiuse tree, which belongs to the family of Euphorbiaceae [1]. It is a drought-tolerant small tree or large shrub with vegetative and generative growth during the rainy season [2] and it has the capacity to adapt to a wide range of soil and agronomic management conditions [3]. Jatropha is a vigorous, drought- and pest-tolerant plant making it interesting for climate change mitigation because it contributes to carbon sequestration [4]. Its oil represents a source for efficient biofuel production [5,6].

Successful cultivation of jatropha requires understanding of agronomic management from the early stage to its yielding phase. Its agronomic performance depends on the genotype, the environmental conditions, and the interaction between these two factors [7]. Multi environmental testing conditions are needed to assess the effects of these three factors in order to select the best performing cultivars [8].

The right establishment method, plant spacing and planting configuration for jatropha depend on environmental conditions [9,10]. To increase the survival rate at the establishment phase, jatropha seedlings are traditionally raised in nurseries for a period of several weeks before the crop is established by transplanting to the field [11]. The use of nurseries evokes economical questions. During its early age, narrow spacing of 2 m × 2 m led to more branches, bigger stems, smaller canopy diameters, and less seed yield per hectare compared to wide spacing of 3 m × 3 m or 4 m × 4 m [4]. Others recommended a general plant spacing of 2 m × 2 m, which is to be adapted to 3 m × 2 m or 3 × 1.5 m in

regions with higher mean temperature [12]. This is in agreement with findings that relate the optimal spacing to the aridity index or sum of rainfalls of the respective site [11].

The objectives of this study were to assess the (1) effects of varying plant spacings, crop establishment methods and genotypes on seed yield, seed quality and plant height and (2) discuss agronomic management strategies for jatropha.

## 2. Materials and Methods

### 2.1. Locations

The field experiments were situated at Ihosy (Ihorombe Region, Madagascar, 2014–2017) and Batchenga (Centre Region, Cameroon, 2013–2016), which have different edapho-climatic characteristics (Table 1). Batchenga had a humid climate with mean annual rainfall of 1375 mm and mean temperature of 29.6 °C during the trial period of 3 years. Ihosy had a dry-subhumid climate with mean annual rainfall of 635 mm and mean temperature of 24.9 °C during the trial period of 3 years. Detailed weather data is provided in Tables S1 and S2. Typically, the abundance of inflorescences follows a unimodal pattern at Ihosy and a bi-modal pattern at Batchenga; at least 50% of the jatropha plants have inflorescences from October to March at Ihosy, whereas at least 50% of the jatropha plants have inflorescences from April to July and again from September to December at Batchenga. Batchenga has a sandy loam soil and Ihosy has a loamy sand soil. Both locations were characterized by low pH values, as well as low (Ihosy) and very low (Batchenga) amounts of available phosphorus in the soil according to Food and Agriculture Organization (FAO) soil fertility classification [13]. Locally available mineral fertilizers and plant protectants were applied according to location and plant development specific requirements. Weed management was performed mechanically and with herbicides.

**Table 1.** Characterization of the locations and field experiments.

| Parameter | Batchenga | Ihosy |
|---|---|---|
| Climate classification [1] | As | Cwa |
| Aridity Index [2] | 1.33 | 0.50 |
| Altitude (m) | 457 | 715 |
| Annual average rainfall (mm) | 1375 | 635 |
| Average annual temperature (°C) | 29.6 | 24.9 |
| Seasonal flowering pattern [3] | 13 weeks (April–July) + 14 weeks (September–December) | 22 weeks (October–March) |
| Soil type | Sandy loam | Loamy sand |
| pH | 5.0 | 5.6 |
| $C_{org}$ (%) | 1.53 | 0.75 |
| $N_{total}$ (%) | 0.16 | 0.06 |
| $C_{org}/N_{total}$ | 9.6 | 12.5 |
| P (mg/kg) | 1.3 | 6.3 |
| Ca (mg/kg) | 1664 | 553 |
| Mg (mg/kg) | 211 | 147 |
| K (mg/kg) | 165 | 328 |
| Crop establishment methods | Transplanting of nursery raised plants | Transplanting of nursery raised plants, direct sowing |
| Plant spacings | 1.5 m × 4 m, 2.0 m × 4 m | 1.5 m × 4 m, 2.0 m × 4 m |

**Table 1.** *Cont.*

| Parameter | Batchenga | Ihosy |
|---|---|---|
| Number of genotypes | 5 | 5 |
| Number of replications | 3 | 2 |
| Number of plants per observation unit (= single plot) | 48 | 48 |
| Number of plants in total | 1440 | 1915 |

[1] Koeppen–Geiger climate classification [14]: As = Equatorial summer dry Climate, Cwa = Warm temperate, dry winter and hot summer. [2] Long-term estimates using LocClim software (Food and Agriculture Organization of the UN, FAO/SDRN, Rome, Italy) [15]. [3] Period with at least 50% of plants having inflorescences.

### 2.2. Agronomy

In Ihosy, seeds were sown in perforated polybags of 1.5 liters, which contained substrate composed of 1/3 local topsoil, 1/3 sand, and 1/3 cow dung. Seedlings were raised in a nursery under shade for 60 days and under plain sunlight for rustification for additional 14 days prior to transplanting to the field.

In Batchenga, seeds were sown in polybags, which contained substrate composed of 2/3 local topsoil and 1/3 organic manure. Seedlings were raised in a nursery under shade for 114 days and a rustification period without shade of 14 days prior to transplanting to the field. At nurseries at both locations, watering was done complementing rainfall in order to keep the substrate moist. Seedlings were transplanted in pits of 30 cm × 30 cm × 30 cm and fertilized with 30 g/pit of NPK 11/22/16 in Ihosy and 30 g/pit of NPK 20/10/10 in Batchenga.

The direct sowing variant was tested only at Ihosy. The soil was plowed and seeds were sown in pits of 30 cm × 30 cm × 30 cm, containing local topsoil, 1200 g of cow dung and 30 g of NPK 11/22/16. Two seeds were sown at a depth of 2 cm and the weaker seedling in each pit was rogued after 74 days. Direct sowing in the field was done one week after sowing of seeds in the nursery with the onset of rains.

Plants were arranged in spacings of 1.5 m × 4 m and 2.0 m × 4 m corresponding to planting densities of 1667 and 1250 plants/ha, respectively, at both locations.

The jatropha genotypes for this study were selected from the breeding program of JatroSolutions (JatroSolutions GmbH, Stuttgart, Germany). At Batchenga, the plant material under study was half sib progeny descending from jatropha varieties JSPE101, JSPE102, and JSPE002 and experimental jatropha varieties JSPE10d and JSPE10x, and, at Ihosy the plant material under study was half sib progeny descending from jatropha varieties JSPE101, JSPE102, and JSPE002 and experimental jatropha varieties JSPE10d and JSPE00x. In previous field experiments, JSPE101 and JSPE102 had shown high and stable yields across years and locations. JSPE002 and JSPE00x produce seeds lacking phorbol esters and are considered edible jatropha variants. JSPE101, JSPE102, and JSPE002 were registered in the Paraguayan national register of protected cultivars in 2018 [16].

### 2.3. Data Collection

Data was collected plot-wise with single plots comprising 48 plants. In total, 1915 plants were observed at Ihosy and 1440 plants at Batchenga. Fruit and seed yields were measured monthly during each season. Plant height was scored at the end of the trial period. Representative samples of 90 cm$^3$ of seeds were drawn from each plot at the end of each season to examine seed oil content, single seed mass and seed bulk density. Seed oil content was determined with the seed analyzer "minispec mq7.5" (Bruker Optik GmbH, Rheinstetten, Germany) based on nuclear magnetic resonance (NMR).

## 2.4. Experimental Design and Statistical Analysis

The experiment at Ihosy was laid out as a strip-split-plot experiment with two replications. Plant spacings and crop establishment methods represented the main strip plots and genotypes were assigned to sub plots within the spacings-establishment method combinations.

Analysis of variances for the field experiment at Ihosy was carried out on each annual data set using the following model

$$y_{ijkl} = \mu + s_i + m_j + sm_{ij} + g_k + sg_{ik} + mg_{jk} + smg_{ijk} + r_l + sr_{il} + mr_{jl} + smr_{ijl} + \varepsilon_{ijkl} \tag{1}$$

where $\mu$ is the general mean, $s_i$ is the effect of the spacing, $m_j$ is the effect of the establishing method, $g_k$ is the genotype effect, $sm_{ij}$, $sg_{ik}$, $mg_{jk}$, and $smg_{ijk}$ are the corresponding interaction effects among the main effects, $r_l$ is the effect of the replication, $sr_{il}$ is the main plot error for the spacing, $mr_{jl}$ is the main plot error for the establishing method, $smr_{ijl}$ is the interaction error and $\varepsilon_{ijkl}$ is the residual error term.

The experiment at Batchenga was laid out as a split-plot experiment with three replications. The different spacings represented the main plots and genotypes were assigned to sub plots.

Analysis of variances for the field experiment at Batchenga was carried out on each annual data set using the following model

$$y_{ikl} = \mu + s_i + g_k + sg_{ik} + r_l + sr_{il} + \varepsilon_{ikl} \tag{2}$$

where $\mu$ is the general mean, $s_i$ is the effect of the spacing, $g_k$ is the genotype effect, $sg_{ik}$ is the interaction effect among the main effects, $r_l$ is the effect of the replication, $sr_{il}$ is the main plot error for the spacing and $\varepsilon_{ikl}$ is the residual error term. For the annual analyses per location, the ANOVA directive of Genstat 19 (VSN International, Hemel Hempstead, UK) [17] was used and multiple comparisons of means were carried out after Bonferroni adjustment of the significance level.

For the combined analyses of plant spacings and genotypes in both field experiments across years and locations, the following mixed model was used

$$y_{iklmn} = \mu + yl_{mn} + s_i + syl_{imn} + g_k + gyl_{kmn} + gs_{ik} + sgyl_{ikmn} + ryl_{lmn} + sryl_{ilmn} + \varepsilon_{iklmn} \tag{3}$$

where $\mu$ is the general mean, $yl_{mn}$ is the fixed effect of the environment comprising location and year, $s_i$ is the fixed effect of the spacing, $g_k$ is the fixed genotype effect and $gs_{ik}$, $syl_{imn}$, $gyl_{kmn}$, and $sgyl_{ikmn}$ are the corresponding interaction effects among the main effects, $ryl_{lmn}$ is the random effect of the replication per environment, $sryl_{ilmn}$ is the random main plot error for the spacing per environment and $\varepsilon_{iklmn}$ is the random residual error term. Due to the repeated measurements nature in the analysis of perennial crops, it is recommended to account for serial correlation of measurements taken on the same observational unit [18]. Therefore, we applied a first order autoregressive (AR1) model structure on the residual error term $\varepsilon_{iklmn}$. The combined analysis across environments was carried out using the VCOMPONENTS and REML directives in Genstat 19.

## 3. Results and Discussion

### 3.1. Effects of Locations and Years

The combined analyses of plant spacings and genotypes showed significant differences between the environments for all traits (Table 2). Jatropha is a perennial tree crop, which may start flowering and fruiting within one year from planting. The amounts of seeds harvested within the first two years of cultivation were negligible at both locations (<180 kg per ha and year, Figure 1) and augmented considerably in the third year. Similar observations were reported in earlier studies [11]. Neither of the two testing sites was characterized by fertile soil conditions. Though, interestingly, the seed yields at Ihosy (2.2 t/ha on average in the set common genotypes, Table 3) were significantly higher than those at Batchenga (0.9 t/ha, Table 4), although the latter has a humid climate. Batchenga is

characterized by two short growing seasons per year, which might be too short for most genotypes to complete their reproductive cycle and produce higher seed yields. Plants at the humid site had reached a significantly higher plant height (2.8 m, Table 4) than at the dry-subhumid site (1.6 m, Table 3), which is in agreement with earlier observations [4,8]. It is likely that jatropha plants react with an increased vegetative growth and biomass accumulation at humid sites like Batchenga, whereas generative growth and seed production is fostered at less humid sites like Ihosy. This observation could lead to a new understanding of jatropha cultivation and deserves further research across more locations and years.

**Table 2.** P values of fixed effects from combined mixed model analyses across environments (DF = degrees of freedom).

| Source | DF | Seed Yield Per Plant | Seed Yield Per Hectare | Oil Content | Single Seed Mass | Seed Bulk Density | Plant Height |
|---|---|---|---|---|---|---|---|
| Environment | 5 | <0.001 ** | <0.001 ** | <0.001 ** | <0.001 ** | <0.001 ** | <0.001 ** |
| Spacing | 1 | 0.368 | <0.001 ** | 0.959 | 0.175 | 0.066 | 0.481 |
| Genotype | 3 | <0.001 ** | <0.001 ** | <0.001 ** | <0.001 ** | <0.001 ** | <0.001 ** |
| Spacing × Genotype | 3 | 0.641 | 0.831 | 0.168 | 0.192 | 0.157 | 0.645 |
| Spacing × Environment | 5 | 0.430 | <0.001 ** | 0.027 * | 0.049 * | 0.003 ** | 0.378 |
| Genotype × Environment | 15 | <0.001** | <0.001 ** | <0.001 ** | <0.001 ** | <0.001 ** | <0.001 ** |
| Spacing × Genotype × Environment | 15 | 0.454 | 0.365 | 0.070 | 0.680 | 0.857 | 0.784 |

* significant at $p \le 0.05$, ** significant at $p \le 0.01$.

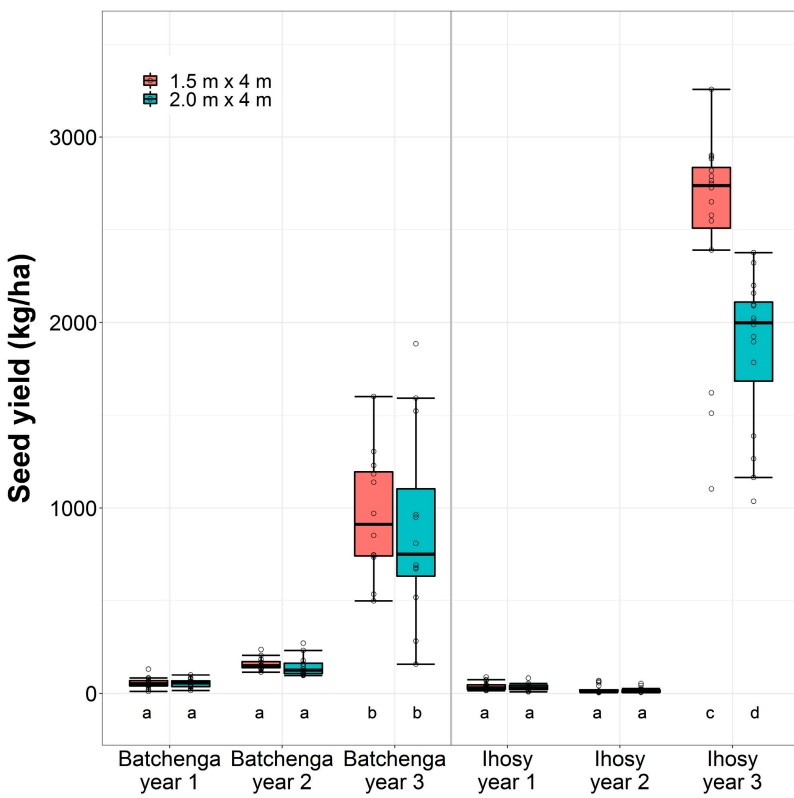

**Figure 1.** Boxplots of seed yields (kg/ha) per location, growing year and spacing variant. Means of treatments with the same letter are not significantly different at $\alpha = 5\%$.

**Table 3.** Means ± standard errors at Ihosy for spacings averaged over all half sib families (in bold) and per half sib family in the corresponding spacing (3rd year). Different capital and small letters, respectively, indicate significant differences at α = 5%.

| Treatment | Seed Yield (g/plant) | Seed Yield (kg/ha) | Oil Content (%) | Single Seed Mass (g) | Seed Bulk Density (kg/L) | Plant Height (cm) |
|---|---|---|---|---|---|---|
| **1.5 m × 4 m** | **1501 ± 79 A** | **2501 ± 131 A** | **42.9 ± 0.4 A** | **0.61 ± 0.01 A** | **0.62 ± 0.02 A** | **162 ± 4 A** |
| JSPE101 (1.5 m × 4 m) | 1780 ± 59 b | 2965 ± 99 e | 43.8 ± 0.3 a | 0.53 ± 0.02 d | 0.53 ± 0.02 b | 155 ± 7 a |
| JSPE10d (1.5 m × 4 m) | 994 ± 162 a | 1657 ± 269 ab | 39.4 ± 0.6 b | 0.57 ± 0.01 cd | 0.56 ± 0.01 ab | 152 ± 6 a |
| JSPE102 (1.5 m × 4 m) | 1616 ± 29 b | 2692 ± 48 de | 44.0 ± 0.3 a | 0.59 ± 0.01 bcd | 0.60 ± 0.01 ab | 184 ± 12 a |
| JSPE002 (1.5 m × 4 m) | 1640 ± 43 b | 2733 ± 71 de | 43.6 ± 0.2 a | 0.67 ± 0.01 a | 0.67 ± 0.01 a | 162 ± 6 a |
| JSPE00x (1.5 m × 4 m) | 1475 ± 205 b | 2457 ± 341 cde | 43.8 ± 0.2 a | 0.67 ± 0.02 a | 0.67 ± 0.02 a | 156 ± 1 a |
| **2.0 m × 4 m** | **1487 ± 67 A** | **1859 ± 83 B** | **42.8 ± 0.5 A** | **0.59 ± 0.01 A** | **0.56 ± 0.02 B** | **168 ± 4 A** |
| JSPE101 (2.0 m × 4 m) | 1642 ± 89 b | 2052 ± 111 bcd | 44.2 ± 0.3 a | 0.52 ± 0.01 d | 0.52 ± 0.01 b | 169 ± 9 a |
| JSPE10d (2.0 m × 4 m) | 971 ± 60 a | 1214 ± 75 a | 38.9 ± 0.7 b | 0.55 ± 0.02 d | 0.54 ± 0.004 b | 165 ± 9 a |
| JSPE102 (2.0 m × 4 m) | 1593 ± 47 b | 1991 ± 59 bcd | 43.5 ± 0.2 a | 0.58 ± 0.003 bcd | 0.58 ± 0.004 ab | 168 ± 15 a |
| JSPE002 (2.0 m × 4 m) | 1738 ± 62 b | 2173 ± 77 bcd | 43.6 ± 0.1 a | 0.64 ± 0.02 abc | 0.64 ± ab | 168 ± 8 a |
| JSPE00x (2.0 m × 4 m) | 1492 ± 43 b | 1865 ± 54 abc | 44.0 ± 0.1 a | 0.65 ± 0.01 ab | 0.64 ± ab | 168 ± 13 a |

**Table 4.** Means ± standard errors at Batchenga for spacings averaged over all half sib families (in bold) and per half sib family in the corresponding spacing (3rd year). Different capital and small letters, respectively, indicate significant differences at α = 5%.

| Treatment | Seed Yield (g/plant) | Seed Yield (kg/ha) | Oil Content (%) | Single Seed Mass (g) | Seed Bulk Density (kg/L) | Plant Height (cm) |
|---|---|---|---|---|---|---|
| **1.5 m × 4 m** | **560 ± 52 A** | **933 ± 86 A** | **34.9 ± 0.3 A** | **0.68 ± 0.01 A** | **0.44 ± 0.003 A** | **280 ± 6 A** |
| JSPE101 (1.5 m × 4 m) | 677 ± 48 ab | 1128 ± 79 ab | 33.8 ± 0.4 a | 0.62 ± 0.01 a | 0.43 ± 0.004 a | 286 ± 10ab |
| JSPE10d (1.5 m × 4 m) | 417 ± 62 ab | 695 ± 104 ab | 34.7 ± 0.2 ab | 0.69 ± 0.01 a | 0.43 ± 0.002 a | 251 ± 2 ab |
| JSPE102 (1.5 m × 4 m) | 809 ± 81 ab | 1348 ± 135 b | 35.5 ± 0.5 ab | 0.70 ± 0.01 a | 0.44 ± 0.011 a | 301 ± 15 b |
| JSPE002 (1.5 m × 4 m) | 405 ± 42 ab | 676 ± 70 ab | 36.2 ± 0.4 ab | 0.72 ± 0.02 a | 0.45 ± 0.002 a | 274 ± 8 ab |
| JSPE10x (1.5 m × 4 m) | 491 ± 127 ab | 819 ± 212 ab | 34.2 ± 0.4 ab | 0.68 ± 0.02 a | 0.43 ± 0.011 a | 287 ± 8 ab |
| **2.0 m × 4 m** | **690 ± 99 A** | **863 ± 123 A** | **35.1 ± 0.3 A** | **0.69 ± 0.02 A** | **0.43 ± 0.004 A** | **273 ± 7 A** |
| JSPE101 (2.0 m × 4 m) | 937 ± 292 ab | 1172 ± 365 ab | 35.1 ± 0.8 ab | 0.64 ± 0.04 a | 0.43 ± 0.005 a | 280 ± 13 ab |
| JSPE10d (2.0 m × 4 m) | 359 ± 122 a | 449 ± 152 a | 34.6 ± 0.7 ab | 0.66 ± 0.01 a | 0.42 ± 0.007 a | 237 ± 2 a |
| JSPE102 (2.0 m × 4 m) | 1047 ± 200 b | 1308 ± 250 ab | 34.7 ± 0.6 ab | 0.70 ± 0.03 a | 0.43 ± 0.005 a | 297 ± 2 b |
| JSPE002 (2.0 m × 4 m) | 517 ± 159 ab | 646 ± 198 ab | 36.7 ± 0.4 b | 0.75 ± 0.04 a | 0.46 ± 0.005 a | 263 ± 15 ab |
| JSPE10x (2.0 m × 4 m) | 590 ± 84 ab | 738 ± 106 ab | 34.5 ± 0.8 ab | 0.69 ± 0.03 a | 0.43 ± 0.005 a | 288 ± 11 ab |

In the present study, the average single seed mass was significantly higher in seeds harvested at the humid site (0.69 g in the set of common genotypes grown at both locations (JSPE101, JSPE10d, JSPE102, and JSPE002)) than that of seeds harvested at the dry-subhumid site (0.58 g in the set of common genotypes). This implies that more flowers were produced that turned into fruits at the latter site. On the other hand, oil content in seeds harvested at the dry-subhumid site (42.6% in the set of common genotypes) was significantly higher than that in seeds at the humid site (35.2% in the set of common genotypes). Previous studies indicated a strong influence of the environment on seed yields and oil content [7] and a relation of higher protein content in seeds harvested in humid environments to increased nitrogen uptake due to better water availability [19]. The results of the present study further underpin the need for further research on the environmental factors that determine seed and oil yield in jatropha.

## *3.2. Effects of Spacing*

The seed yields per unit area and the seed bulk density in the narrow spacing variant at Ihosy were significantly higher than those in the wider spacing variant (Tables 3 and 5), whereas there was no effect at Batchenga (Tables 4 and 6). Correspondingly, there was a significant spacing-by-environment interaction effect, when data was analyzed across locations (Table 2, Figure 2a).

**Table 5.** P values of effects from analysis of variance at Ihosy (3rd year, DF = degrees of freedom).

| Source | DF | Seed Yield Per Plant | Seed Yield Per Hectare | Oil Content | Single Seed Mass | Seed Bulk Density | Plant Height |
|---|---|---|---|---|---|---|---|
| Spacing | 1 | 0.836 | 0.026 * | 0.687 | 0.141 | 0.040 * | 0.517 |
| Establishing method | 1 | 0.453 | 0.466 | 0.637 | 0.712 | 0.573 | 0.635 |
| Establishing method × Spacing | 1 | 0.494 | 0.434 | 0.599 | 0.460 | 0.250 | 0.932 |
| Genotype | 4 | <0.001** | <0.001 ** | <0.001 ** | <0.001** | 0.001 ** | 0.447 |
| Genotype × Establishing method | 4 | 0.425 | 0.368 | 0.225 | 0.127 | 0.073 | 0.657 |
| Genotype × Spacing | 4 | 0.690 | 0.420 | 0.609 | 0.972 | 0.854 | 0.493 |
| Genotype × Establishing method × Spacing | 4 | 0.308 | 0.277 | 0.819 | 0.604 | 0.762 | 0.552 |

* significant at $p \leq 0.05$, ** significant at $p \leq 0.01$.

**Table 6.** P values of effects from analysis of variances at Batchenga (3rd year, DF = degrees of freedom).

| Source | DF | Seed Yield Per Plant | Seed Yield Per Hectare | Oil Content | Single Seed Mass | Seed Bulk Density | Plant Height |
|---|---|---|---|---|---|---|---|
| Spacing | 1 | 0.313 | 0.581 | 0.589 | 0.849 | 0.542 | 0.125 |
| Genotype | 4 | <0.001 ** | <0.001 ** | 0.002 ** | 0.001 ** | 0.013 * | <0.001 ** |
| Genotype × Spacing | 4 | 0.659 | 0.911 | 0.288 | 0.541 | 0.328 | 0.944 |

* significant at $p \leq 0.05$, ** significant at $p \leq 0.01$.

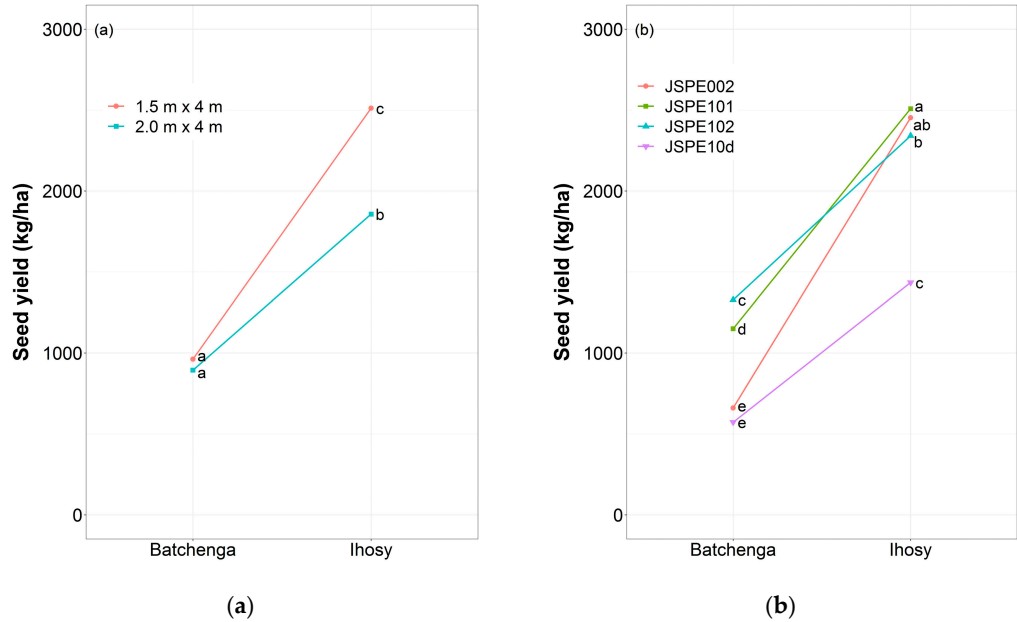

**Figure 2.** Mean seed yields (kg/ha, 3rd year) of spacings (**a**) and half sib families (**b**) at Batchenga and Ihosy displaying spacing-by-environment and genotype-by-environment interactions, respectively. Means followed by same letter are not significantly different at α = 5%.

Spacing in tree crops like jatropha is a compromise, as the arrangement of the trees should allow mechanization of weeding using machinery as well as efficient canopy management, and, at the same time, make best use of the available land to increase seed yields per unit area. In the present study, we compared wide (2.0 m) and narrow (1.5 m) spacing of trees within rows, and we had a distance between rows of 4 m, which according to our experience is suitable for managing weeds with tractors and attached machinery for at least three years before pruning should be done. Analysis across environments indicated significantly higher seed yields per hectare in the narrow spacing variant at the dry-subhumid site, whereas no effect was measurable at the humid site. Interestingly, spacing had no effect on seed yields per single plant in neither of the locations. This suggests that narrow spacing can further contribute to increase yields and better exploit the available area particularly in productive environments. A further increase of the plant density per hectare at locations, where vegetative growth apparently is favored, might lead to a marginal increase in seed yield per hectare, too. However, strong plant biomass growth in combination with narrowly spaced trees will complicate canopy management and fruit harvest, favor the development of fungal diseases and promote competition between plants for sunlight and other resources [4]. Therefore, especially for humid environments, varieties that are tolerant to high planting densities are urgently needed to allow for a better exploitation of the available land. Similarly, for promising environments, it will also be necessary to find the optimal spacing, which allows to obtain the maximum seed yields per unit area under realistic field conditions (canopy management, fungal diseases). Canopy management is an important aspect in older jatropha stands, which can be addressed by partial plant elimination and/or pruning [4,20], depending on agro-ecological conditions and plant architecture. Pruning, however, can lead to plant stress and therefore to a reduction of productivity [7,21]. In the long run, varieties with reduced plant height or canopy diameter will be developed that should facilitate harvesting and canopy management practices.

*3.3. Effects of Establishment Method*

The establishment method at Ihosy had no significant effects on seed yields or any other assessed parameter (Table 5). Nevertheless, all half sib families exhibited slightly higher seed yields after direct sowing (Figure 3). Furthermore, survival rates in the third year of growth were similar with the exception of JSPE10d, which had higher survival rate in the direct sowing variant (95% in comparison

to 85% survival rate, respectively; data not shown). These results encourage to re-think the common practice of raising seedlings in a nursery before transplanting to the field.

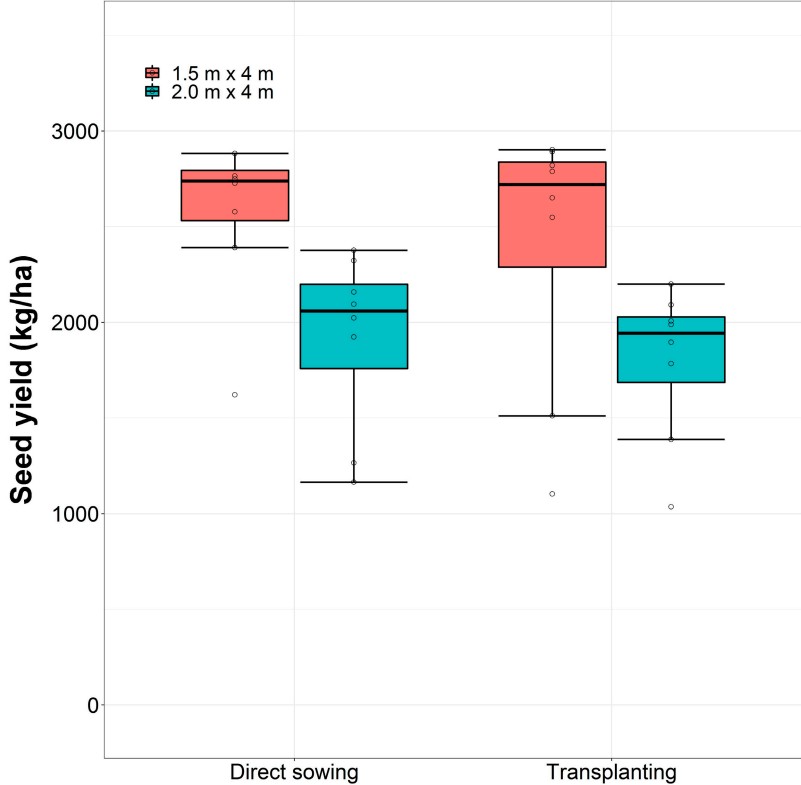

**Figure 3.** Box plots displaying seed yields (kg/ha) per spacing and establishing method at Ihosy (3rd year). The least significant difference at the 5% error level for pairwise comparisons between spacing after identical crop establishing methods is 637 kg/ha, otherwise 1293 kg/ha.

Similarly, it was shown that jatropha plants established through direct sowing had a slightly better survival rate after one year than plants grown from seedlings raised in a nursery and a significantly better survival than plants from stem cuttings [22]. In another study, a vigorous tap root was found in plants established through direct sowing, whereas plants from seedlings raised in polybags had abnormal tap roots and cuttings completely lacked such a structure [23]. Plants with a more vigorous root system have better capacity of water and nutrient uptake from deeper soil layers. Therefore, direct sowing could become an interesting alternative establishment method especially in jatropha growing areas with low aridity indices. However, in comparison to a nursery system, one should be aware of the risk of considerable additional labor cost during the establishment phase (1) for weed management until the seedlings are able to compete with other plants for water and nutrients and (2) for irrigation particularly in the absence of rainfall. In addition, the use of high-quality seeds with high germination capacity and strong vigor becomes more important with direct sowing.

In summary, direct sowing bears the risk of plant losses due to drought, competition with weeds and damage by pests and diseases during the first months, because young plants are more susceptible to stress factors than older plants. However, if field conditions can be efficiently mastered to avoid plant losses at an early stage, direct sowing of seeds might be a more resource efficient approach instead with the potential of significantly better seed yields and survival rates over the years. Further research will be needed to determine, if and under which circumstances potentially higher management costs during the establishment phase (in comparison to a nursery system) can be compensated by higher returns in later years.

*3.4. Effects of Genotypes*

Analyses across environments (Table 2) and per location (Tables 5 and 6) revealed highly significant genotype effects of the different half sib families for all traits under study (except for plant height at Ihosy). This demonstrates the importance of the correct choice of variety to make jatropha projects profitable. There was no significant interaction between genotypes and spacing at neither of the locations indicating broad adaptation to different spacings.

A recent study demonstrated great potential for new jatropha hybrid varieties, which exhibited high yields per unit area and less genotype-by-environment interaction [24]. Genotype-by-environment interaction describes a phenomenon, where genotypes perform differently in varying environments. In the present study, we found highly significant genotype-by-environment interaction effects for all measured parameters (Table 2) and it can be nicely seen for seed yield per hectare in Figure 2b. While the half sib progeny of JSPE002 had relatively low yields and was not significantly different from the weakest half sib progeny of JSPE10d at Batchenga, the same progeny had very good yields and was not different from the best performing half sib progeny of JSPE101 at Ihosy. The best performing genotypes (half sib progeny of JSPE101 and JSPE102) also exhibited a rank change, but their yields were nevertheless relatively stable and on a relatively high level at both locations. Plant breeders often select genotypes with broad adaptation to varying environments and the latter two represent good candidates for use as parents in jatropha hybrid breeding.

## 4. Conclusions

The outcomes of this study underpin the necessity to adjust the agronomic management (such as plant spacing and establishment method) from planting onwards, depending on the particular growing location and selected variety. The highest seed yield in the present study was realized at the dry-subhumid location (Ihosy) by the half sib progeny of JSPE101 in a spacing of 1.5 m × 4 m with a mean seed yield of 3.0 t/ha in the third year of growth. Many jatropha projects in the past had failed to reach similar productivity and consequently collapsed. Many aspects of agronomic practices in jatropha cultivation however remain unclear and deserve further elucidation to allow future jatropha projects to turn into success stories.

**Supplementary Materials:** The following are available online at http://www.mdpi.com/2073-4395/9/8/465/s1, Table S1: Temperature and rainfall parameters at Batchenga during the trial period, Table S2: Temperature and rainfall parameters at Ihosy during the trial period.

**Author Contributions:** Conceptualization, M.M. and E.S.; methodology, M.M.; software, M.M.; validation, Z.T.A., E.D., E.S.; formal analysis, M.M. and Z.T.A.; investigation, Z.T.A. and E.D.; resources, Z.T.A. and E.D.; data curation, E.S.; writing—original draft preparation, Z.T.A.; writing—review and editing, E.S., Z.T.A., E.D. and M.M.; visualization, M.M.; supervision, M.M.; project administration, M.M.

**Funding:** This research received no external funding.

**Acknowledgments:** We are grateful to all technicians involved in the management of field experiments and in data collection.

**Conflicts of Interest:** The authors declare no conflict of interest.

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
