# Peer review of "Effects of Genotype, Direct Sowing and Plant Spacing on Field Performance of Jatropha curcas L."

_agronomy, doi:10.3390/agronomy9080465_

Round 1

Reviewer 1 Report

Comments

This manuscript describes comparative analysis of the field performance of five Jatropha varieties in semi-arid (Ihosy, Madagascar) and humid tropics (Batchenga, Cameroon). Two different between-row spacing practices, and different planting methods were also tested in a considerably large scale experiment, offering an important and valuable information on the effects of environments, agronomic managements, and genotypes and the their combinations on the field performance of Jatropha. Datasets in the manuscript appear sound, and discussion in the text seems adequate in general. However, there are a couple of minor issues to be addressed, as described below.

Table 1:

A parameter “Seasonal flowering pattern (weeks)” is not self-explanatory, and should offer more explanation on the definition.

Only the “Annual average rainfall” and “Average annual temperature” of the locations during the trial period of 3 years are shown in the Table 1 and the text. However, data on the variations of these parameters between years, and their monthly fluctuations, should be very important and informative to discuss differential field performance of Jatropha in these locations. These information can be added in supplementary materials.

Tables 3 and 4,

Page 7, lines 161-163:

In these tables, it is confusing that names of genotypes are written in the column “Treatment”. Within the tables, there are two lines in bold letters starting from “1.5 m x 4 m” and ”2.0 m x 4 m”, but no genotypes are indicated. I presume these data represent overall average, but explanation is missing. These points should be revised.

In the first paragraph in page 7, it is described that “the average single seed mass” was higher “at the humid site (0.69 g in the set of common genotypes) than that of seeds harvested at the dry-subhumid site (0.58 g)”. In this phrase, definition of “the set of common genotypes” is unclear and should be revised. Does this correspond to the data in bold font in Tables 3 and 4? In this case, the latter value “0.58 g” is missing in the Table 3. These points should be revised.

Reviewer 2 Report

The article is well-written and topic important. Jatropha is a promising multipurpose shrub or tree with possibly great potential in biofuel production, and it certainly makes sense to explore its optimal growing conditions. My only concerns are the low replicate numbers (two versus three per treatment) and slight differences in growing conditions between the two locations (Madagascar and Cameroon) that in addition to the genotype, planting mode and plant spacing may have influenced results. For instance, there was a 4.8-fold difference in the soil P content between Ihosy and Batchenga. Authors however claim that both locations are characterized by low amounts of P in the soil; this might require some clarification (I am not a nutrient specialist). And why were different types of NPK fertilizer applied in the two locations? Table 3 needs clarification; explain the different letters (a, b, c and d). I still consider the paper as a useful contribution to our understanding of jatropha growing conditions, and the recommendations made by authors seem very sensible.
